SOFTWARE

# Optical Neuroimage Studio (OptiNiSt): Intuitive, scalable, extendable framework for optical neuroimage data analysis

Yukako Yamane[1]*, Yuzhe Li[1,2], Keita Matsumoto[3], Ryota Kanai[3], Miles Desforges[1,3], Carlos Enrique Gutierrez[1,4], Kenji Doya[1]*

**1** Neural Computation Unit, Okinawa Institute of Science and Technology Graduate University, Okinawa, Japan, **2** Mathmatical Informatics Lab, Nara Institute of Science and Technology, Nara, Japan, **3** Araya Inc., Tokyo, Japan, **4** Beyond AI Promotion Division, SoftBank Corp., Tokyo, Japan

* yukako.yamane@oist.jp (YY); doya@oist.jp (KD)

## Abstract

Advancements in calcium indicators and optical techniques have made optical neural recording common in neuroscience. As data volumes grow, streamlining the analysis pipelines for image preprocessing, signal extraction, and subsequent neural activity analyses becomes essential. Challenges in analysis includes 1) ensuring data quality of original and processed data at each step, 2) selecting optimal algorithms and their parameters from numerous options, each with its own pros and cons, by implementing or installing them manually, 3) systematically recording each analysis step for reproducibility, and 4) adopting standard data formats for data sharing and meta-analyses. To address these challenges, we developed Optical Neuroimage Studio (OptiNiSt), a scalable, extendable, and reproducible framework for creating calcium data analysis pipelines. OptiNiSt includes the following features. 1) Researchers can easily create analysis pipelines by selecting multiple processing modules, tuning their parameters, and visualizing the results at each step through a graphic user interface in a web browser. 2) In addition to pre-installed tools, new analysis algorithms can be easily added. 3) Once a processing pipeline is designed, the entire workflow with its modules and parameters are stored in a YAML file, which makes the pipeline reproducible and deployable on high-performance computing clusters. 4) OptiNiSt can read image data in a variety of file formats and store the analysis results in NWB (Neurodata Without Borders), a standard data format for data sharing. We expect that this framework will be helpful in standardizing optical neural data analysis protocols.

## Author summary

Streamlining data analysis workflows in optical neural recording is essential as data volumes increase. Challenges include ensuring data quality, selecting

**Data availability statement:** The code for the software framework is available on GitHub at https://github.com/oist/optinist. The sample data and example workflows are available on Zenodo at https://doi.org/10.5281/zenodo.13357960.

**Funding:** This work was funded by the program for Brain/MINDS and Brain/MINDS 2.0 from Japan Agency for Medical Research and Development (AMED) (https://www.amed.go.jp/en/index.html) under JP19dm0207001, JP23wm0625001 to K.D. The funders had no role in study design, data collection and analysis, decision to publish, or preparation of the manuscript.

**Competing interests:** The authors have declared that no competing interests exist.

optimal algorithms for analyses, and maintaining reproducibility. To address these, we developed Optical Neuroimage Studio (OptiNiSt), a framework for creating scalable, reproducible calcium imaging analysis pipelines. OptiNiSt allows users to design analysis workflows by selecting processing modules, tuning parameters, and visualizing results through an intuitive web-based interface. In addition to pre-installed analysis tools, new algorithms can be easily added. The stored entire analysis pipeline ensures reproducibility and enable deployment on high-performance computing clusters. This framework aims to facilitate the standardization of data analysis, improving reproducibility, and collaboration within the neuroscience community.

## Introduction

Given the increasing adoption of two-photon, light-sheet, and endoscopic microscopes, huge volumes of optical imaging data are being collected. Simultaneously, various image processing and cell detection software tools have been developed and are available in open source [1–4]. Additionally, a wide variety of downstream analyses for cell populations are available in open source. However, results often vary depending on the algorithm used, particularly for tasks such as cell detection or spike inference, where obtaining ground truth data is impractical [5]. Misclassifying or neglecting the neural signals might affect experimental results [6–8]. Determining which algorithm is preferable for a given dataset is challenging. Ideally, multiple algorithms should be tested and compared [9]. However, difference in versions of operating system, programming language and libraries, required data formats, and output visualization formats complicate such comparisons, often requiring extensive preparation and coding. Additionally, the application of some algorithms needs extensive computation resources and high-performance computer (HPC) clusters to obtain results within a reasonable amount of time [10].

Therefore, we developed Optical Neuroimage Studio (OptiNiSt) to help researchers quickly test multiple analytical methods, interactively visualize results, and build reproducible data analysis pipelines for efficient processing on HPC clusters. OptiNiSt allows users to test multiple analytical methods with different parameters without coding or setting up computing environments and to add custom analysis modules. The pipelines can be saved in YAML format coherently with the analysis results for better reproducibility. OptiNiSt aims to make a wide variety of analytical methods accessible to experimental researchers, to promote the use of new analytical methods proposed, and at the same time, to promote the standardization of analysis protocols. OptiNiSt's saving format is compliant with Neurodata Without Borders (NWB) standards [11], resulting in widespread usage because of the diligence in maintaining the community. OptiNiSt is free, open source (GNU General Public License v3.0), and available from GitHub and DockerHub.

Existing tools like CIAtah [12] can compare different cell detection algorithms but requires MATLAB licenses and lacks full pipeline records. Mesmerize [13], although

Python-based and open source, focuses mainly on CaImAn [3] operations. NeuroCAAS [14] is a cloud-based service for neuroscience data analyses, which eliminates the burden of configuring hardware and software dependencies and promotes reproducible analysis. OptiNiSt is designed to allow users to easily and interactively explore various analyses through a GUI frontend on a web browser, while its computing backend can be run either on the local machine or a remote server. This allows data analysis beginners to explore various methods without coding and to reach large-scale analyses on HPC clusters or cloud servers.

To achieve Findable, Accessible, Interoperable, and Reusable (FAIR) data [15], sharing not only the data but also the software used for processing is essential for reproducing research findings [16]. While open science principles are largely accepted, it is practically difficult for researchers to prioritize research data and software management over other processing demands [9]. OptiNiSt allows easily saving analysis pipelines and results to enhance reproducibility and collaboration, and standardization of neural data analysis protocols. It can also benefit computational researchers by providing novel methods as plug-in modules for increased accessibility and adoption.

## Design and implementation

OptiNiSt was developed with four major design principles: 1) intuitive GUI, 2) extendable to include new modules, 3) reproducible and scalable to be deployed on HPC clusters, and 4) adoption of standard formats for interoperability. The users interact with OptiNiSt through a web browser as the frontend. OptiNiSt has three interfaces. WORKFLOW is for creating analysis pipelines, VISUALIZE is for visualizing plots of the analysis results, and RECORD is for managing the created pipelines. Through GUI, the user can create and run the analysis pipelines and inspect the results. All the information necessary to reproduce the pipeline, including the path to the original data and parameters for analyses, is saved in a folder with a specific ID. The list of pipelines is easily recognized and reproduced through the RECORD interface. The system is also designed to transfer pipelines to HPC clusters for script execution. The analysis pipelines are saved in the YAML format, and the results, in the NWB format. Users with modest experience in coding can add their own analytical tools as modules written in Python. The installation procedure is described in S1 Text.

### Architecture

The backend uses Snakemake (https://snakemake.github.io; [17]) to control Python environments and execution flow of scripts, enabling the implementation of different analyses that require different environments to be used in the same workflow. React (https://react.dev) and FastAPI (https://fastapi.tiangolo.com) were adopted as the web library and framework for the user interface, making it a cross-platform application that functions regardless of the operating system (Fig 1).

### GUI for workflow design

OptiNiSt adopts a graphic interface akin to MATLAB's Simulink or LabVIEW, allowing users to construct pipelines by connecting nodes representing operations. These pipelines allow branching, thereby facilitating complex processes, and enable complete reproducibility by generating unique IDs for each operation. Such specifications enable users to create highly adaptable workflows tailored to their needs. This design also permits users to simultaneously test multiple algorithms and their parameters within the same workflow, facilitating comparison and enabling a fairer selection of algorithms.

On the WORKFLOW page of the GUI, a single operation module is expressed as a rectangular node (Fig 2A). The connectors of a node indicate the function's input or output, determining the data type by their colors. The connector type pops out when the cursor is close to it so that the user can easily confirm. The parameters are displayed by clicking the PARAM button on the node, and can be easily modified in the GUI (Fig 2B).

The nodes are placed and connected intuitively by clicking and dragging. The rule ensuring that connectors of the same color can only be connected to each other prevents confusing the data type. The first input data can be multi-page images (.tiff) as raw imaging data, tabular data (.csv), metadata (.nwb or.h5), or MATLAB data (.mat) for other types of

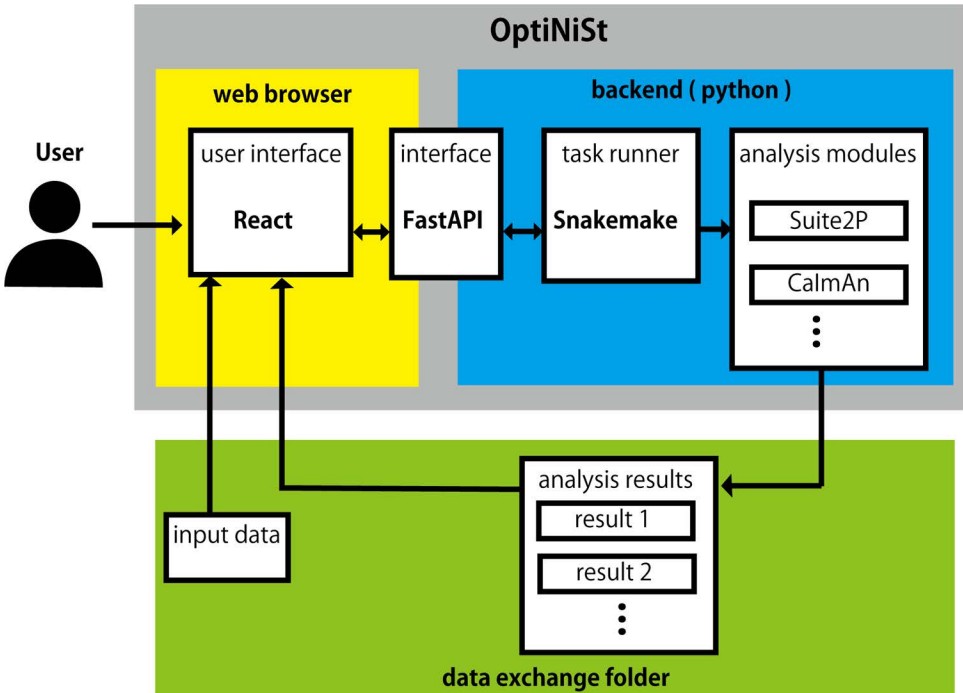

**Fig 1. Architecture of OptiNiSt. Users interact with OptiNiSt through a web browser.** According to the user's input, it creates the SnakeMake script that controls the flow of Python scripts and runs them. The results are inspected continuously and updated, thus making them visible from the browser.

data, such as behavioral data or fluorescence change values of the cells. The variation in accepted datasets makes the possible pipelines flexible. A user can start with motion correction and region of interest (ROI) detection algorithms or an analysis of the population activity in already identified cells. The GUI includes a node to import data acquired from commonly used microscopes. Therefore, it is also possible to directly input raw data acquired from microscopes (Inscopix [.isxd], NIKON [.nd2], and Olympus [.oir]).

To run the workflow, the user clicks the run button (S1 Movie). An indicator shows whether the process was completed with or without errors, and in case of errors, the error messages are visible by clicking on the indicator. All processing results are saved in a format that is compliant with the Optophysiology module of NWB. Metadata, such as subject IDs, data acquisition framerates, and fluorescence types, can also be input via the GUI. This feature simplifies the sometimes tedious creation of.nwb files for the purpose of generating public data. The workflow created by OptiNiSt is saved in a YAML format and can be reproduced on the working field by one click. Fig 2C shows an example of a workflow comparing multiple cell detection algorithms. Individual algorithms need a specific Python environment, but these environments are selected automatically while running the workflow. Upstream analyses of imaging data, such as motion correction or ROI detection, often take time. OptiNiSt has the functionality to check each step of the pipeline and execute calculations from newly added nodes or nodes with changed parameters, thereby automatically avoiding unnecessary repetition of calculations.

Currently, the cell detection algorithms implemented in OptiNiSt are Suite2P [2], CalmAn [3], and LCCD [4]. Pre-installed analytical tools are listed in S1 Table. Users can also add custom modules by coding with Python.

### GUI for visualization

The VISUALIZE page enables quick image checks and result visualization with multiple plots and comparisons. One troublesome part in calcium imaging data analysis is checking whether the algorithm appropriately captures the cell

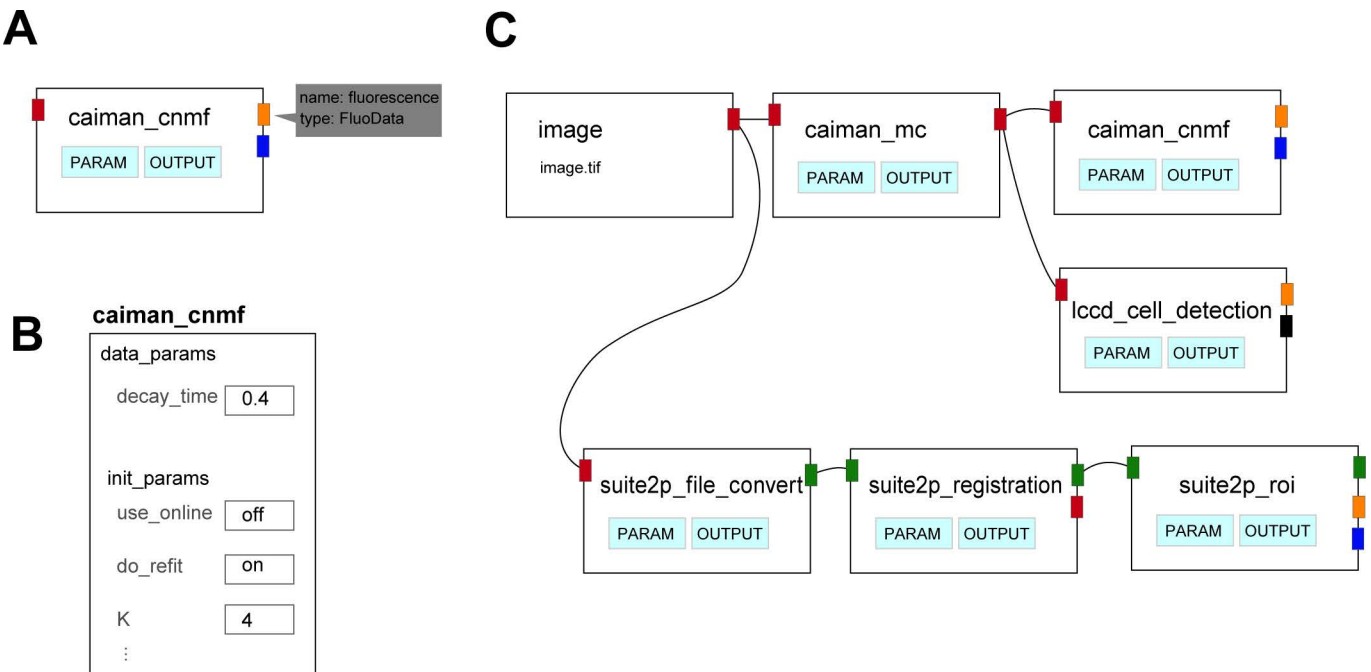

**Fig 2. Graphical user interface for workflow.** A: An example of a node as a unit of operation.The node is defined with an input (a small connector on the left side, shown in red in this example) and outputs (connectors on the right side, shown in yellow and blue). The red input connector indicates image data (vertical pixels x horizontal pixels x frames), the yellow output connector, fluorescence data (number of cells x time) and the blue output connector, iscell data (number of cells). The data type of each connector is displayed by hovering the cursor over it. In this example, the data type of the yellow connector is shown. Each node has two buttons (PARAM and OUTPUT), allowing easy parameter modification and visualization of computation results with a single click. B: An example of a panel for modifying parameters. The panel is displayed by clicking the PARAM button on the node (A). C: A pipeline can be created by connecting nodes. In this example, image movie data is analyzed using two types of motion correction and three types of cell-detection algorithms.

areas. Fig 3A and 3B shows the panels for visualizing the cell area and corresponding fluorescence. These two panels are linked, making it easy to match cells on the map with traces (Supporting Information S2 Movie). In addition to the functionality of showing the detected cell areas, OptiNiSt allows for manually manipulating the cell area by merging, deleting, and adding new candidate cell areas and their time course (Supporting Information S2 Movie) and the manipulation information is saved in an NWB file. Input images can be displayed as a time-lapse video, where the user selects the data and frame sequence (Supporting Information S3 Movie).

Downstream analysis applied to the multiple-cell activities might need other plotting styles. Graphs, such as bar, scattergrams, and pseudo-colored images, have already been implemented. Fig 3C-E shows example plot types. These are plot for correlation (C) and PCA (contribution of individual neuron (D) and projection of sample points onto PC1 and PC2 (E)).

## GUI for workflow management

Data analysis often involves iterative adjustment, making pipeline complex. To organize pipelines, OptiNiSt provides the RECORD page (Fig 4), which lists created pipelines, assigning unique IDs, and allows naming and sorting by date, time, and duration to minimize confusion.

Listed pipelines can be reproduced with one click, rerun on the WORKFLOW page, and reviewed. Workflow details, including modules used and error status, are also available. Results and pipelines can be saved locally, enabling easy

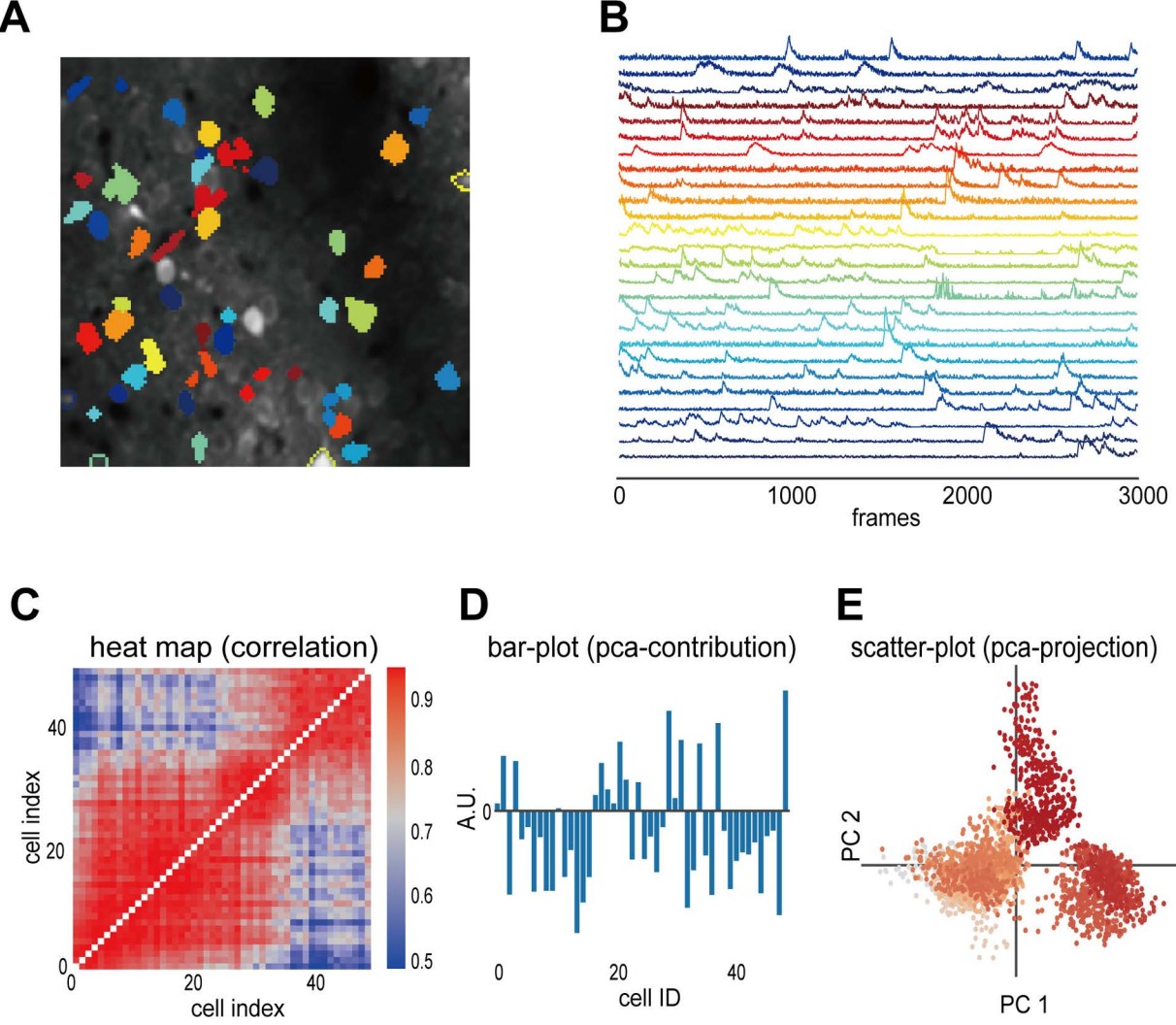

**Fig 3. Graphical user interface for visualization.** A: An example plot visualizing the detected cell areas. B: The corresponding fluorescence time course for each cell shown in **A.** The cell areas and fluorescence time courses are linked; clicking on a cell displays its fluorescence time course (see Supporting Information S2 Movie). C-E: Examples of implemented plot types for visualizing analysis results. C: A heat map displaying the pairwise correlation of cellular activity. D: A bar graph showing the contributions obtained by PCA on the neural population activity. E: A scatter plot showing the projection of the activity onto PC1 and PC2 of PCA on the neural population activity.

sharing. This feature provides a convenient option for the public release of analysis pipelines, which is often a requirement of journals.

## Saving format

OptiNiSt describes the details of the workflow in YAML format, which is crucial for pipeline reproducibility. From this file, it is possible to generate a Snakemake script, which is required for actual computations (also in YAML format). The computation results are documented in HDF files compliant with NWB. For motion correction results, where the image file sizes are large, users can choose whether to save them, providing flexibility. Therefore, with these files, the workflow and computation results are fully documented.

**A**

| Timestamp | ID | Name | Success | Reproduce | SnakeFile | NWB | Delete |
|---|---|---|---|---|---|---|---|
| 2025-01-01 12:00:00 | abcdefgh | my_caiman | ✓ | ↩ | ⬇ | ⬇ | ✕ |
| 2025-01-01 12:10:00 | 12345678 | my_s2p | ✓ | ↩ | ⬇ | ⬇ | ✕ |
| 2025-01-01 12:15:00 | abcd1234 | my_lccd | ✓ | ↩ | ⬇ | ⬇ | ✕ |
| ⋮ | ⋮ | ⋮ | ⋮ | ⋮ | ⋮ | ⋮ | ⋮ |

**B**

### Details

| Function | nodeID | Success | NWB |
|---|---|---|---|
| data.tiff | input_0 | ✓ | ⬇ |
| caiman_mc | caiman_mc_efgh5678 | ✓ | ⬇ |
| caiman_cnmf | caiman_cnmf_ijklm1234 | ✓ | ⬇ |
| ⋮ | ⋮ | ⋮ | ⋮ |

**Fig 4. Graphical user interface for workflow management.** A: A table designed to mimic the appearance of the RECORD field. When a pipeline is created, it is automatically registered in the list. The ID is unique to each pipeline. The name is assigned by the user. When the pipeline is completed without errors, a check is placed in the Success column. By clicking the Reproduce button, the pipeline is recreated on the WORKFLOW page. Snakefiles and NWB files can also be downloaded with one click. A pipeline is made up of multiple functions. B: A table mimicking the appearance of the details view. Detailed information for each pipeline, such as the name of each node and the status of computations is also available.

### Implementation of custom-made functions

OptiNiSt allows users to add custom analysis functions as modules (Fig 5). Users only need to add a few files: 1) the main function file (Fig 5A), 2) an initialization file (Fig 5B), 3) the YAML file for parameters and default values (Fig 5C) and 4) an environment file (Fig 5D). Additionally, to enable GUI, the information on the new function module should be added to the existing initialization file (Fig 5E).

### Script execution

OptiNiSt provides an option to execute the workflow designed by GUI on computing servers by downloading the Snakemake file, which includes all the information about the workflow, from the RECORD page. Input and output paths are changeable by environmental variables for OptiNiSt. The script, run_cluster.py, executes the workflow with an option assigning the Snakemake config file path. This procedure simply runs backend processes and is convenient for executing pipelines for multiple large datasets on HPC or clusters.

**new_algo**: name of new algorithm or group of functions
**new_script**: name of script
**function1**: name of individual function
**function1_GUI**: name of individual function appears on the GUI

## A

studio/app/optinist/wrappers/**new_algo**_wrapper/**new_script**.py

```
from optinist.api.dataclass.dataclass import  *

def function1(
                your_input_data_name: your_input_data_type, ...,
                outputdir: str,
                params: dict= None,
                **kwargs,
            ) -> dict(your_output_data_name=your_output_data_type):

    import your_necessary_python_module...

    main part of new function

    info = {
    "your_output_data_name": your_output_data_fromat(your_output_data, file_name="your_output_data"),
    ...,

    }

    return info
```

## B

/optinist/wrappers/**new_algo**_wrapper/__init__.py

```
from .new_script import function1

new_algo_wrapper_dict = {
  'new_algo': {
     function1_GUI: {
        'function': function1,
        'conda_name': 'new_algo_conda_env_name',
     }
   }
}
```

## C

studio/app/optinist/wrappers/params/ **function1_GUI**.yaml

```
new_param_category_1:
  new_param1: your_default value_for_new_param1
  new_param2: your_default value_for_new_param2
new_param_category_2:
  new_param3:  your_default value_for_new_param3
  ...
```

## D

studio/app/optinist/wrappers/conda/**new_algo**.yaml

```
dependencies:
   - python=3.9
   -pip
   -pip:
           -xxx
           -yyy
```

## E

studio/app/optinist/wrappers/__init__.py

```
from .xxxx_wrapper import xxxx_wrapper_dict
from .yyyy_wrapper import yyyy_wrapper_dict
...
from .new_algo_wrapper import new_algo_wrapper_dict

wrapper_dict = {}
wrapper_dict.update(**xxxx_wrapper_dict)
wrapper_dict.update(**yyyy_wrapper_dict)
...
wrapper_dict.update(**new_algo_wrapper_dict)
```

**Fig 5. Adding a new function module.** Files should be placed in the appropriate place, which is shown on the top of each panel. A: The main function determining the calculation should be written in a fixed format, enabling easy input and output configuration. Green letters indicate the lines that the user defines. B: The initialization file for the module. C-D: The parameters and conda environments are defined in YAML style. E: To make the module appear in the graphical user interface, information on the new function module should be added to the existing initialization file.

## Results

Here we present examples of building workflows and visualizing the analysis results using sample 2-photon imaging data obtained from the parietal area of a mouse during auditory stimulation from 12 speakers surrounding the animal [18]. The details of the data and instruction for reproducing the results are described in S1 and S2 Text.

### Comparison of motion correction algorithms

The first step of imaging data processing is motion correction. The pipeline shown in Fig 6 compares motion correction algorithms: CaImAn (Norm Corre: [19]), TurboReg [20], and Suite2P rigid and nonrigid registrations. Fig 6A shows the workflow, where branching from the image node enables input to each algorithm. OptiNiSt seamlessly handles such branching processes by running each node in an isolated environment. To compare parameter settings, two suite2p_ registration-nodes are created with rigid or nonrigid method separately selected. Clicking on the PARAMS-button of a node shows the list of adjustable parameters (S4 Movie). Clicking on OUTPUT in each node displays the mean image

PLOS Computational Biology

**A**

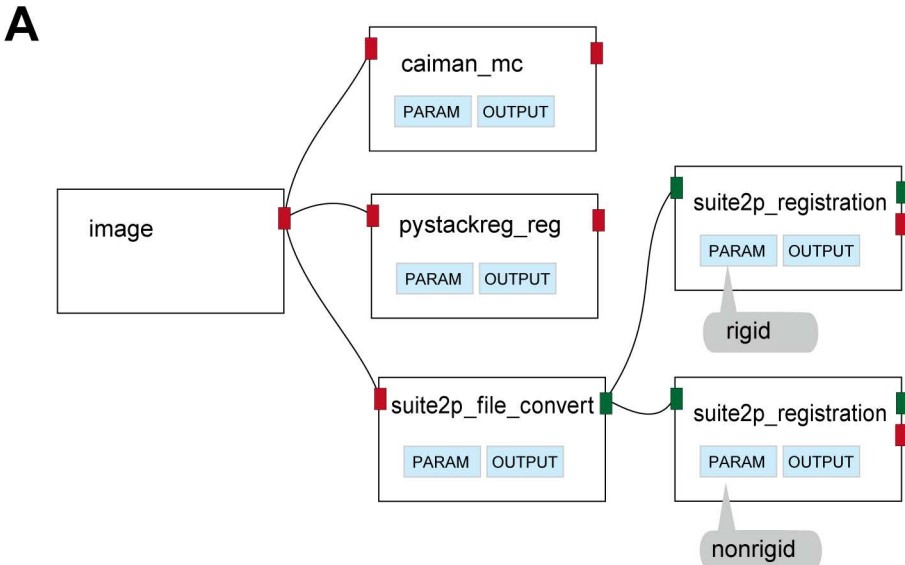

**B**

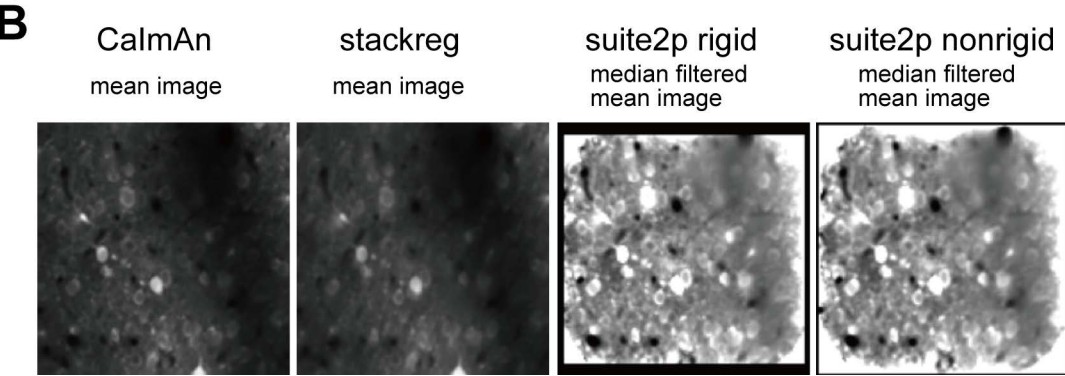

**Fig 6. Example of motion correction comparison.** A: The pipeline to compare different motion correction algorithms, B: The mean (across frames) images after applying individual motion correction algorithms. For suite2P, median filtered mean images are shown.

after performing motion correction (Fig 6B), enabling quick comparison. The current possible scenario for comparison is, first manually create the pipelines for rough comparison of algorithms and parameters in an interactive way with GUI, then, if extended parameter search is needed, create a list of snakemake-files to run in CUI.

## Comparison of cell detection algorithms

Fig 7 compares cell detection algorithms using the same 2-photon data as Fig 6. In this example, we compared cell extraction algorithms: CaImAn cnmf [3], Suite2P [2], and LCCD [4]. The number of cells extracted by each algorithm was 47, 55, and 33, respectively with numbers varying by parameter settings. In this example, mostly the default parameters were used, but adjustments are possible via GUI (S1 and S4 Movies). Fig 7A shows the pipeline for comparison. Motion correction by CaImAn's NormCorre, then, the output was split for each detection algorithm. The resulting cell areas (Fig 7B) and corresponding fluorescence traces (Fig 7C) are shown on the VISUALIZE page. Linking the cell ROI and fluorescence panels allow easy verification of correspondence. The fluorescence trace of cells indicated by the white arrows in Fig 7B is shown as the upper

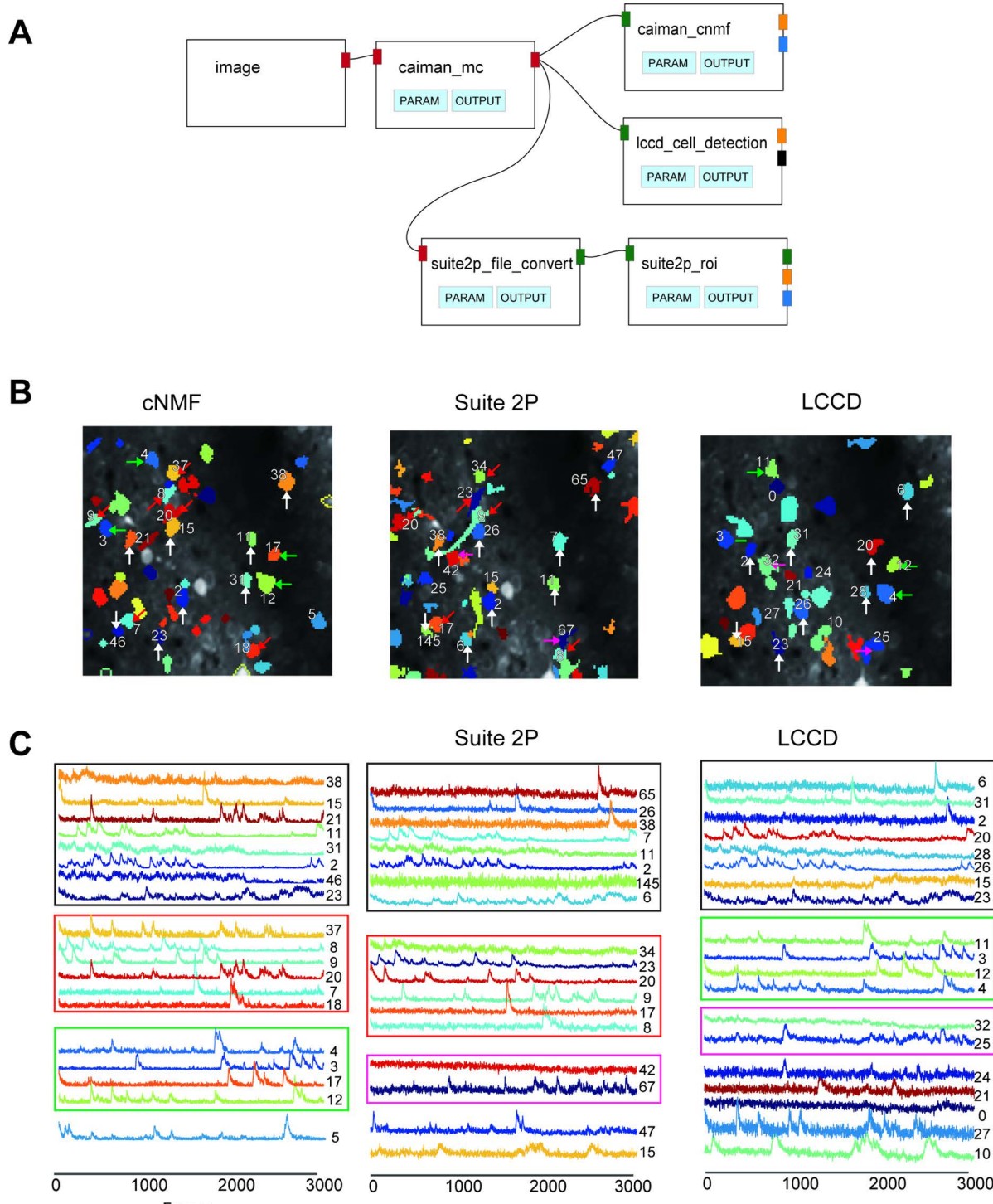

**Fig 7. Example of cell detection comparison.** A: The pipeline to compare different cell detection algorithms. B: Cell areas detected by three different algorithms. Three example cells detected by all the algorithms, as indicated by white arrows. Red, green, and magenta arrows indicate examples of cells detected by both cNMF and Suite2P, cNMF and LCCD, or Suite2P and LCCD, respectively. C: The fluorescence time course of detected cells. The top eight traces indicated by black rectangles correspond to the cells indicated by the white arrows in B. Colors of the rectangles indicates the cells with the arrow of the same color.

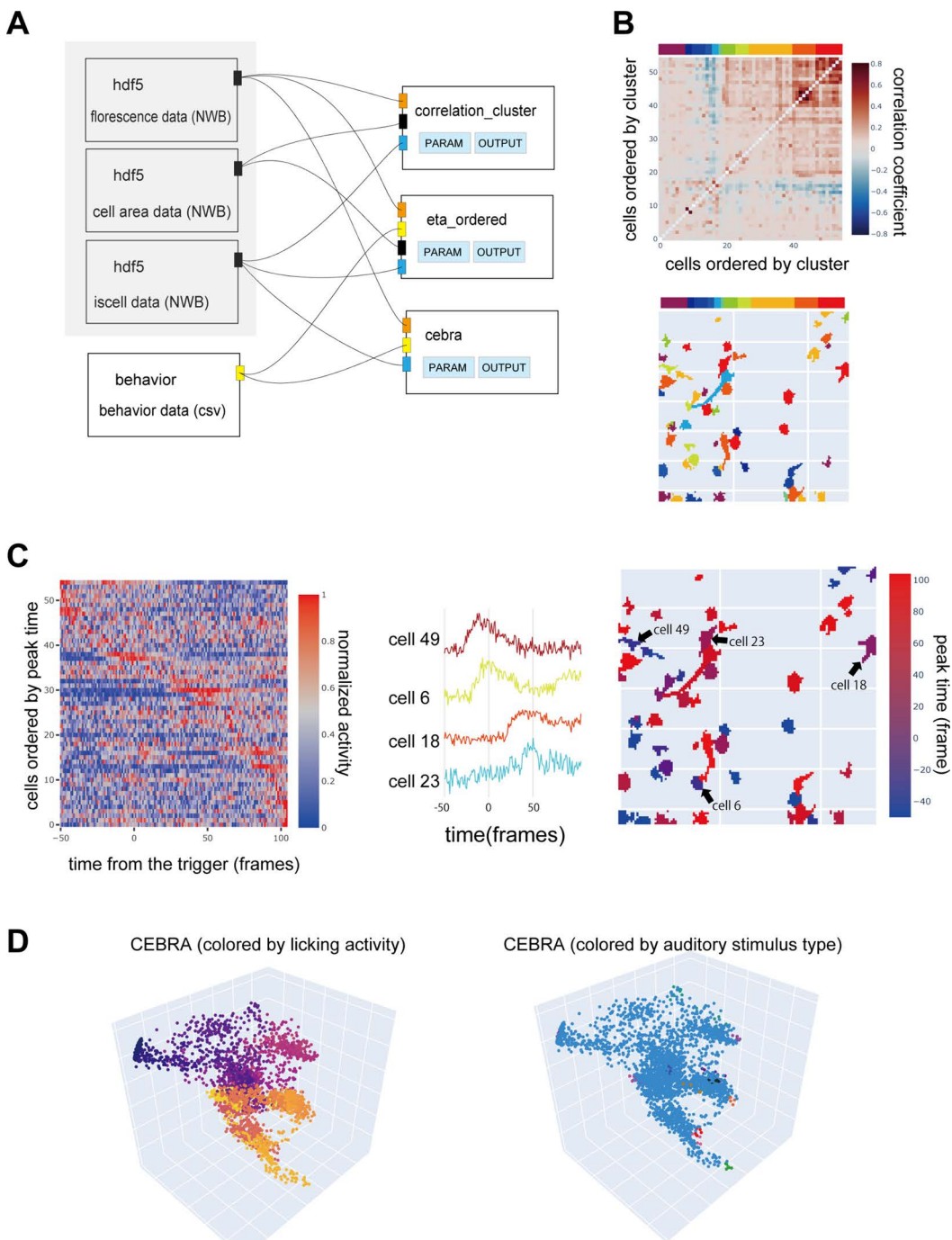

**Fig 8. Example of downstream analysis.** A: The pipeline for performing pairwise correlation analysis, triggered averaging, and CEBRA. The three input nodes in the shaded area reference the same NWB (HDF5) file. These nods refer to different datasets: fluorescence, cell area, and iscell, respectively. The outputs from these nodes connect as inputs to three downstream analysis nodes. B: Resulting plot of correlation analysis. Correlation matrix ordered by hierarchical clustering with threshold of 0.5 (upper) and the cell areas (lower) are shown. The colored bar on the top indicates the cluster. C: Resulting plot of event triggered averaging. Left: Stimulus-triggered averaging of mean fluorescence of all cells across trials. Center: Mean traces of example cells showing earlier and later increases in activity. Right: Cell areas are indicated and colored based on the peak time. The cells indicated by the arrows are the same as those in the center trace. D: Latent values identified using multiple neuronal activity, stimulus, and licking information using CEBRA. The colors of the images are based on licking (left) and stimulus type (right), respectively.

eight traces in Fig 7C (black rectangles). These cells were detected across all three algorithms, but the traces were not the same; specifically, the cells No 38, 21 and 46 of cNMF looked quite different from those of the other two.

### Neural population data analyses

OptiNiSt supports downstream analyses of neural population activity, as exemplified in Fig 8. Here, the output of suite2p (fluorescence, iscell, and cell position), which was saved in the previous example (Fig 7) as a.nwb file, was fed into three different analytical tools: pairwise correlation, event-triggered averaging, and CEBRA [21] (Fig 8A). Suite2P detected 266 ROIs, qualifying 55 cells. Information on whether an ROI has passed the qualification (iscell), the fluorescence of individual cell, and data on cell areas are saved in single.nwb file and the three different input modules were used to assign each of them.

The module, "correlation_cluster," calculates the pairwise Pearson's correlation coefficients between all neurons and clusters based on hierarchical clustering. The ordered correlation matrix (Fig 8B, upper) shows two strongly correlated clusters (red, orange). The positions of these functionally clustered cells can be examined as shown in the lower panel of Fig 8B, which depicts cluster identity by color.

The "eta_ordered" node calculates the event-triggered averaging of the fluorescence. The trigger time is based on the behavior data, which are imported from a.csv file. The data length was matched to the number of frames. In this example, the trigger is the timing of auditory stimulus onset. Fig 8C (left) shows the activity, with some cells showing a peak around frames 0 and 50. Examples of these cells are shown in the middle panel. The position of these cells is shown in the right panels, together with the peak time indicated by different colors.

Combining simple analyses is crucial for data quality checks and response trends but can be tedious to code from scratch. In OptiNiSt, it is possible to embed specific calculations for calcium imaging into nodes. These nodes allow customization in terms of saving values to NWB and drawing figures with relatively minimal coding.

To showcase advanced node creation, a node for "CEBRA" [21], which finds latent space based on multiple neural activity and behavioral parameters, was introduced. As this example is for demonstration, we only show the latent space discovered by supervised learning. In this example, the licking behavior and the auditory stimulus timing from 12 different positions, as well as the fluorescence of multiple cells, were fed into the node to find the 10-dimensional latent activity. The figure shows three arbitrarily chosen dimensions with behavioral parameters as color codes.

### Performance in large-scale datasets

Motion correction and cell detection may require significant memory size and computation time, depending on the specific algorithms used and availability of GPUs. The OptiNiSt framework itself does not have any constraints for the execution of large datasets. As an example, it has been confirmed that the modules pre-installed in OptiNiSt can analyze 12 GB of imaging data in one go using a Desktop PC (Mac Pro, OS: Ventura, memory size: 192 GB). Additionally, OptiNiSt incorporates both the multi-session cNMF, which performs cell detection on multiple small datasets, and the online cNMF, which iteratively analyzes data while loading them.

## Availability and future directions

### Availability

OptiNiSt is open-source and available on GitHub (https://github.com/oist/optinist). Users can install OptiNiSt to Linux, Mac, and Windows machines by the Python "pip install" command or by downloading a virtual machine from DockerHub (oist-ncu/optinist). The user guide, documentation, and tutorials are provided at https://optinist.readthedocs.io/en/latest/. There are Slack user communities available to discuss the use of OptiNiSt, and there is a possibility of discussion on GitHub. All the information about the discussion, including instruction on how to join and FAQs, is available at https://github.com/oist/optinist or https://optinist.readthedocs.io/en/latest/.

## Limitations

The types of plots currently supported are limited to cell-area visualizations required for cell detection and basic plots such as line plots, bar plots, scatter plots, and heat maps. Other plot types require custom user-defined functions. Supporting interactive plots for more types would improve user-friendliness. Currently, when performing repetitive tasks, such as grid searches for parameter optimization or running the pipeline independently on multiple datasets, users must manually adjust parameters or datasets one by one via the GUI or manually edit the Snakefile. Enabling such tasks to be performed more efficiently through the GUI would significantly improve usability.

## Future directions

The future direction includes expanding analysis modules by enabling researchers to implement their methods as OptiNiSt modules and encouraging analytical method developers to contribute. Furthermore, lazy loading, which loads only a portion of the data into memory for computations such as image data visualization and memory-intensive motion correction, has not been implemented. However, it is necessary to implement it to support efficient large-scale data visualization and processing.

Enabling OptiNiSt pipelines to run on NeuroCAAS would enhance its user-friendliness, allowing both local and remote execution on powerful cloud hardware. Another extension is applying the OptiNiSt framework to other physiological data beyond calcium imaging.

We hope that the OptiNiSt framework can promote testing of various analytical methods, contributing to the creation of reproducible analysis pipelines.

## Supporting information

**S1 Table. Pre-installed analytical tools.**
(PDF)

**S1 Text. Software installation instructions and details of the sample data.**
(DOCX)

**S2 Text. Instructions for reproducing the analysis in Figs 6–8.**
(DOCX)

**S1 Movie. Example steps for running a workflow through the GUI.**
(MP4)

**S2 Movie. Example steps for visualizing the cell detection results and interactively examining them.**
(MP4)

**S3 Movie. Example steps for checking the input movie data.**
(MP4)

**S4 Movie. Example steps for running the workflow shown in Fig 6.**
(MP4)

## Acknowledgments

We thank the developers of suite2p, CaImAn, and LCCD for making them open source. We thank Shogo Akiyama, Rei Hashimoto and Yuta Fukuda for their exceptional technical support and development of the software. We appreciate the Scientific Computing and Data Analysis section at OIST for their support, Editage (www.editage.jp) for English editing and Yukiko Yamane for creating the logo. We thank Akihiro Funamizu for providing the sample imaging data.

## Author contributions

**Conceptualization:** Yukako Yamane, Yuzhe Li, Carlos Enrique Gutierrez, Kenji Doya.

**Data curation:** Yukako Yamane.

**Formal analysis:** Yukako Yamane.

**Funding acquisition:** Kenji Doya.

**Project administration:** Keita Matsumoto, Kenji Doya.

**Resources:** Kenji Doya.

**Software:** Yukako Yamane, Yuzhe Li, Keita Matsumoto, Miles Desforges.

**Supervision:** Ryota Kanai.

**Validation:** Yukako Yamane, Yuzhe Li, Keita Matsumoto.

**Visualization:** Yukako Yamane.

**Writing – original draft:** Yukako Yamane.

**Writing – review & editing:** Yuzhe Li, Keita Matsumoto, Ryota Kanai, Miles Desforges, Carlos Enrique Gutierrez, Kenji Doya.

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
