## [Decision Letter · Decision Letter 0]

28 Nov 2024

PCOMPBIOL-D-24-01493Optical Neuroimage Studio (OptiNiSt): intuitive, scalable, extendable framework for optical neuroimage data analysisPLOS Computational BiologyDear Dr. Yamane,

Thank you for submitting your manuscript to PLOS Computational Biology. After careful consideration, we feel that it has merit but does not fully meet PLOS Computational Biology's publication criteria as it currently stands. Therefore, we invite you to submit a revised version of the manuscript that addresses the points raised during the review process.

Please submit your revised manuscript within 30 days Jan 26 2025 11:59PM. If you will need more time than this to complete your revisions, please reply to this message or contact the journal office at ploscompbiol@plos.org. Please include the following items when submitting your revised manuscript: * A rebuttal letter that responds to each point raised by the editor and reviewer(s). You should upload this letter as a separate file labeled 'Response to Reviewers'. This file does not need to include responses to formatting updates and technical items listed in the 'Journal Requirements' section below.

* An unmarked version of your revised paper without tracked changes. You should upload this as a separate file labeled 'Manuscript'. If you would like to make changes to your financial disclosure, competing interests statement, or data availability statement, please make these updates within the submission form at the time of resubmission. Guidelines for resubmitting your figure files are available below the reviewer comments at the end of this letter. We look forward to receiving your revised manuscript. Kind regards, Kim T. Blackwell, V.M.D., Ph.D.Academic EditorPLOS Computational Biology

Hugues Berry

Section EditorPLOS Computational Biology 

Feilim Mac Gabhann

Editor-in-Chief

PLOS Computational Biology

Jason Papin

Editor-in-Chief

PLOS Computational Biology

**Journal Requirements:**

1) Please provide an Author Summary. This should appear in your manuscript between the Abstract (if applicable) and the Introduction, and should be 150-200 words long. The aim should be to make your findings accessible to a wide audience that includes both scientists and non-scientists. Sample summaries can be found on our website under Submission Guidelines:

3) We have noticed that you have a list of Supporting Information legends in your manuscript. However, there are no corresponding files uploaded to the submission. Please upload them as separate files with the item type 'Supporting Information'.

4) We notice that your supplementary information is included in the manuscript file. Please remove them and upload them with the file type 'Supporting Information'. Please ensure that each Supporting Information file has a legend listed in the manuscript after the references list.

5) We have noticed that you have uploaded Supporting Information files, but you have not included a list of legends. Please add a full list of legends for your Supporting Information files after the references list.

Potential Copyright Issues:

- Figure 1. Please confirm whether you drew the images / clip-art within the figure panels by hand. If you did not draw the images, please provide a link to the source of the images or icons and their license / terms of use; or written permission from the copyright holder to publish the images or icons under our CC BY 4.0 license. Alternatively, you may replace the images with open source alternatives. See these open source resources you may use to replace images / clip-art:

- The following Figures contain a logo or branding: Figures: 1, 2, 3, and & 4. We are not permitted to publish this under our CC-BY 4.0 license, even with permission. We ask that you please remove or replace it.

- The following Figures contain screenshots: Figures: 2, 3, and & 4. We are not permitted to publish these under our CC-BY 4.0 license, websites are usually intellectual property and are copyrighted.This includes peripheral graphics of the web browser such as icons and button. We ask that you please remove or replace it.

**Reviewers' comments:**

Reviewer's Responses to Questions

**Comments to the Authors:**

**Please note the review is uploaded as an attachment.**

Reviewer #1: Attached in a separate file.

Reviewer #2: OptiNiSt by Yamane et al is a new framework for creating calcium imaging data analysis pipelines.

I think OptiNiSt is a rather useful addition to the repertoire of platforms/frameworks that exist out there to help neuroscientists (and others) carry out their calcium imaging analysis. I largely agree with the authors that OptiNiSt will help neuroscientists to standardize

OptiNiSt 'packages' some of the standard tools (for cell detection, dimensionality reduction, etc) for calcium imaging analysis and appears to be expandable.

The use of YAML is a good choice and maybe offering a slight advantage over other platforms/frameworks that use JSON files or other options. The fact that OptiNiSt can store analysis results in NWB format is also another potential benefit for its users.

Minor comments:

1. Table 1: I think the paper would greatly benefit by having a figure where the various modules are listed in a nice visual way in catergories (e.g. cell detection, input, dimensionality reduction) instead of this long list in Table 1. Nobody is saying not to keep table 1 as a supplemental file but it is rather difficult to get a visual overview of the modules in the current way that they are presented.

2. Figure 3 I think this figure needs some rearrangement to make the details of Panel A more visible and I think subheadings in panel B and panel C are required to explain to the reader what they are looking at. An alternative would be to move Panel A to supplemental and increase the sizes of panels B and C.

3. Related to Figure 6 and Figure 7. It is not obvious to the reader how and to what extent a user can modify the parameters within individual modules (e.g. different motion algorithms) before running them and after (to optimize the motion correction/cell detection). A discussion of this would be important (I believe there is a single sentence on this on line 309).

4. Figure 8: It is not clear to me whether the plots are interactive. E.g. can you click on the GUI on Figure 8 panel C data and go to the actual projection to visualize the quality of the segmentation and motion correction for example?

5. I suggest that the authors also discuss at the end of the manuscript the limitations of the OptiNiSt.

**Have the authors made all data and (if applicable) computational code underlying the findings in their manuscript fully available?**

Reviewer #1: Yes

Reviewer #2: Yes

PLOS authors have the option to publish the peer review history of their article (what does this mean? ). If published, this will include your full peer review and any attached files.

**Do you want your identity to be public for this peer review?** For information about this choice, including consent withdrawal, please see our Privacy Policy .

Reviewer #1: **Yes: ** Kushal Kolar

Reviewer #2: No

**Figure resubmission:** While revising your submission, please upload your figure files to the Preflight Analysis and Conversion Engine (PACE) digital diagnostic tool, https://pacev2.apexcovantage.com/. PACE helps ensure that figures meet PLOS requirements. To use PACE, you must first register as a user. Registration is free. Then, login and navigate to the UPLOAD tab, where you will find detailed instructions on how to use the tool. If you encounter any issues or have any questions when using PACE, please email PLOS at figures@plos.org. Please note that Supporting Information files do not need this step. If there are other versions of figure files still present in your submission file inventory at resubmission, please replace them with the PACE-processed versions.**Reproducibility:** To enhance the reproducibility of your results, we recommend that authors of applicable studies deposit laboratory protocols in protocols.io, where a protocol can be assigned its own identifier (DOI) such that it can be cited independently in the future. Additionally, PLOS ONE offers an option to publish peer-reviewed clinical study protocols. Read more information on sharing protocols at https://plos.org/protocols?utm_medium=editorial-email&utm_source=authorletters&utm_campaign=protocols

---

## [Decision Letter · Decision Letter 1]

4 Mar 2025

PCOMPBIOL-D-24-01493R1

Optical Neuroimage Studio (OptiNiSt): intuitive, scalable, extendable framework for optical neuroimage data analysis

PLOS Computational Biology

Dear Dr. Yamane,

Thank you for submitting your manuscript to PLOS Computational Biology. After careful consideration, we feel that it has merit but does not fully meet PLOS Computational Biology's publication criteria as it currently stands. Therefore, we invite you to submit a revised version of the manuscript that addresses the few remaining points raised by one of the reviewers.  Also, we would appreciate if you would explain the author contributions if the software tool was indeed written by other developers.

Please submit your revised manuscript within 30 days May 04 2025 11:59PM. If you will need more time than this to complete your revisions, please reply to this message or contact the journal office at ploscompbiol@plos.org. Please include the following items when submitting your revised manuscript:

We look forward to receiving your revised manuscript.

Kind regards,

Kim T. Blackwell, V.M.D., Ph.D.

Academic Editor

PLOS Computational Biology

Hugues Berry

Section Editor

PLOS Computational Biology

**Journal Requirements:**

1) Please upload figure 1 as a separate Figure file in .tif or .eps format. For more information about how to convert and format your figure files please see our guidelines: 

**Reviewers' comments:**

Reviewer's Responses to Questions

**Comments to the Authors:**

**Please note that one of the reviews is uploaded as an attachment.**

Reviewer #1: Attached

Reviewer #2: I would like to thank the authors for considering and implementing my comments and suggestions. I am happy with the status of the revised manuscript.

**Have the authors made all data and (if applicable) computational code underlying the findings in their manuscript fully available?**

Reviewer #1: Yes

Reviewer #2: Yes

PLOS authors have the option to publish the peer review history of their article (what does this mean? ). If published, this will include your full peer review and any attached files.

**Do you want your identity to be public for this peer review?** For information about this choice, including consent withdrawal, please see our Privacy Policy .

Reviewer #1: **Yes: ** Kushal Kolar

Reviewer #2: No

**Figure resubmission:**
---

## [Editor Report · Decision Letter 2]

22 Apr 2025

Dear Yamane,

We are pleased to inform you that your manuscript 'Optical Neuroimage Studio (OptiNiSt): intuitive, scalable, extendable framework for optical neuroimage data analysis' has been provisionally accepted for publication in PLOS Computational Biology.

Best regards,

Kim T. Blackwell, V.M.D., Ph.D.

Academic Editor

PLOS Computational Biology

Hugues Berry

Section Editor

PLOS Computational Biology

---

## [Editor Report · Acceptance letter]

PCOMPBIOL-D-24-01493R2

Optical Neuroimage Studio (OptiNiSt): intuitive, scalable, extendable framework for optical neuroimage data analysis

Dear Dr Yamane,

I am pleased to inform you that your manuscript has been formally accepted for publication in PLOS Computational Biology. Your manuscript is now with our production department and you will be notified of the publication date in due course.

With kind regards,

Livia Horvath
